# One-Pot Synthesized Visible Light-Driven BiOCl/AgCl/BiVO_4_ n-p Heterojunction for Photocatalytic Degradation of Pharmaceutical Pollutants

**DOI:** 10.3390/ma12142297

**Published:** 2019-07-18

**Authors:** Rokhsareh Akbarzadeh, Anvar Asadi, Peter Ozaveshe Oviroh, Tien-Chien Jen

**Affiliations:** 1Mechanical Engineering Science Department, Faculty of Engineering and the Built Environment, University of Johannesburg, Johannesburg 2006, South Africa; 2Research Center for Environmental Determinants of Health (RCEDH), Health Institute, Kermanshah University of Medical Sciences, Kermanshah 67146, Iran

**Keywords:** BiOCl/AgCl/BiVO_4_, photocatalyst, heterojunction, diclofenac, degradation

## Abstract

A novel enhanced visible light absorption BiOCl/AgCl/BiVO_4_ heterojunction of photocatalysts could be obtained through a one-pot hydrothermal method used with two different pH solutions. There was a relationship between synthesis pH and the ratio of BiOCl to BiVO_4_ in XRD planes and their photocatalytic activity. The visible light photocatalytic performances of photocatalysts were evaluated via degradation of diclofenac (DCFF) as a pharmaceutical model pollutant. Furthermore, kinetic studies showed that DCF degradation followed pseudo-first-order kinetics. The photocatalytic degradation rates of BiOCl/AgCl/BiVO_4_ synthesized at pH = 1.2 and pH = 4 for DCF were 72% and 47%, respectively, showing the higher activity of the photocatalyst which was synthesized at a lower pH value. It was concluded that the excellent photocatalytic activity of BiOCl/AgCl/BiVO_4_ is due to the enhanced visible light absorption formation of a heterostructure, which increased the lifetime of photo-produced electron–hole pairs by creating a heterojunction. The influence of pH during synthesis on photocatalytic activity in order to create different phases was investigated. This work suggests that the BiOCl/AgCl/BiVO_4_ p-n heterojunction is more active when the ratio of BiOCl to BiVO_4_ is smaller, and this could be achieved simply by the pH adjustment. This is a promising method of modifying the photocatalyst for the purpose of pollutant degradation under visible light illumination.

## 1. Introduction

Increasing demand for water and energy plays an important role in the selection of sustainable treatment technologies. Photocatalysis with strong oxidizing ability—one of the recent environmentally friendly technologies—has shown the ability to treat different kinds of pollutants, and its role has become more significant courtesy of the concept of solar energy conversion. The widespread use of pharmaceutical products—and consequently their residues in water and wastewater—is a growing concern [1]. If these pollutants are not treated properly, they could be a threat to human health and the environment [2]. Significant efforts have been made in research for designing efficient photocatalysts that can be used for the degradation of these kinds of pollutants [3]. There is lots of successful research on photocatalysts’ synthesis and their application in pollutants degradation. However, in order to be able to use semiconductors commercially, we should be able to utilize the full solar spectrum and therefore must develop photocatalysts, which are active in visible light. Photocatalysis can utilize available solar light for degradation of organic pollutants including pharmaceutical pollutants. Organic pollutant degradation by photocatalysts is either through superoxide radicals or hydroxyl radicals. In recent years, monoclinic bismuth vanadate has attracted certain research interest. The positive attributes of bismuth vanadate (BiVO_4_), including its narrow bandgap of 2.4 eV, inexpensiveness, non-toxicity, abundant availability, and its photocatalytic degradation performance of organic pollutants—especially pharmaceutical pollutants—have made BiVO_4_ a relatively attractive new photocatalyst [4]. However, BiVO_4_ photocatalyst is not efficient enough alone, which might be due to the fast electron–hole recombination rate and insufficient visible light absorption. The construction of a heterojunction is seen to be an effective method of reducing the electron–hole recombination and also improving the region of visible light absorption.

On the other hand, bismuth oxychloride (BiOCl) is a relatively new photocatalyst with stabilized chemical properties including nontoxicity, corrosion resistance, and an open crystalline structure, which has exhibited excellent performance in the photocatalytic degradation of organic pollutants. However, BiOCl like TiO_2_ has a wide bandgap (3.3 eV), which limits its utilization in the visible light range.

Studies indicate that creating a heterojunction between BiOCl and BiVO_4_ can enhance the visible light photo-absorption ability of the photocatalyst as compared to single compounds [5]. Fabrication of a heterojunction composed of two or more semiconductors with the suitable conduction band (CB), valence band (VB), synergistic effect, and oxygen vacancies, promotes the capacity of visible light absorption, which facilitates efficient photogenerated electron–hole pair separation. However, because of the high rate of electron–hole recombination and insufficient visible light absorption, its uses are still limited. 

For the enhancement of both BiVO_4_ and BiOCl photocatalytic activity, different methods such as doping or decorating with metals [6,7] and forming a heterojunction structure [8] have been studied independently. The photocatalytic performance of nBiVO_4_@p-MoS_2_ [9], g-C_3_N4/BiVO_4_ [10], Bi_2_O_3_-BiVO_4_ [11], Co_3_O_4_/BiOCl nanoplates, BiOCl/AgCl composite, and BiOCl microflowers was evaluated, and it was reported that BiOCl- and BiVO_4_-based photocatalysts are effective in organic compound degradation [12,13,14]. Few researchers have studied the ternary BiOCl/BiVO_4_ composites with metals. However, their synthesis methods involved between 2 and 3 steps and higher temperatures were required in their synthesis methods, which can also be complicated [15,16]. A recent publication on ternary g-C_3_N_3_/Ag/AgCl/BiVO_4_ shows that the addition of a transition metal such as silver (Ag) significantly enhances the activity of two semiconductors in a heterojunction [17]. A better electron–hole separation and improved visible light absorption were observed due to the results of charge transfer between the semiconductor and the noble metal [6,18].

In the present work, a novel BiVO_4_/AgCl/BiOCl, n-p heterojunction was synthesized for the first time through a one-pot hydrothermal method. The photocatalytic efficiency of the photocatalysts was evaluated for photocatalytic degradation of diclofenac (DCF).

## 2. Materials and Methods

### 2.1. Material

Bismuth chloride (BiCl_3_), ammonium metavanadate (NH_4_VO_3_), and AgNO_3pa_ of analytical grade were purchased from Sigma-Aldrich, Ltd. (Saint Louis, MO, USA) and used without further treatment.

### 2.2. Preparation of BiOCl/AgCl/BiVO_4_

The BiVO_4_/AgCl/BiOCl at pH 1.2 (BAB1) and BiVO_4_/AgCl/BiOCl at pH 4 (BAB2) were synthesized through a one-pot hydrothermal process. A precursor of 0.32 g of BiCl_3_ was dissolved in 30 mL deionized water and magnetically stirred for 30 min and a white suspension was formed. An amount of 0.11 g of NH_4_VO_3_ was added to this solution and immediately 0.5 mL of 1 M ethanolamine was added. A solution with an original pH of 1.2 was stirred for 30 min and the color changed to pale yellow from orange. In another synthesis, the same procedure was followed, but the pH of the solution was adjusted to 4.0 by 4 M NaOH solution and it was stirred until the pH became stable. Under continuous stirring, 10 mL of AgNO_3_ solution (6 g/L) was added to both solutions which was further stirred for 30 min for the deposition of silver. After stirring for 30 min, the solutions with precipitates were transferred into two Teflon-lined stainless-steel autoclaves with a capacity of 100 mL. The volume of the solutions was brought to 80% of the Teflon flask volume by adding DI water (deionized water). This was mixed thoroughly and heat treated at 180 °C for 4 h. The autoclaves were kept at room temperature to cool down and then the greenish-yellow precipitate was washed and centrifuged with deionized water and ethanol several times to remove organic residuals. Finally, the collected precipitate was dried in an oven at 120 °C for 12 h. The BAB1 and BAB2 were collected and ground in a mortar for further experiments. The effect of pH on the synthesis and photocatalytic activity of BiOCl/AgCl/BiVO_4_ for photocatalytic degradation of DCF was investigated. 

### 2.3. Characterization

The crystal structures of as-prepared photocatalysts were measured by X-ray diffraction (XRD) using a Bruker AXS D8 Advance instrument (Billerica, MA, USA) equipped with a Cu Ka radiation (λ = 0.15406 nm), and were recorded in the range of 10–90° 2θ. The morphology of the heterojunction photocatalysts was investigated by scanning electron microscopy (ZEISS-Auriga Cobra SEM, Oberkochen, Germany) and transmission electron microscopy (JEOL JEM-2010 electron microscopy, Tokyo, Japan). To study the light absorption behavior, Ultraviolet–visible diffuse reflectance spectra (UV-vis DRS) were collected using a UV-vis spectrophotometer (Lambda 950, PerkinElmer, Waltham, MA, USA) at the wavelength range of 200–800 nm. Raman spectroscopy was conducted at room temperature with a WITec alpha300R Confocal Laser Raman Microscope (Ulm, Germany) 633 nm laser in the range of 0–4000 cm^−1^ to provide further information on structure. This is also useful for the investigation of the crystallization and electronic properties of the photocatalyst. The BET-specific surface area and total pore volume of the photocatalysts were measured at 77.35 K by nitrogen adsorption analysis using a Belsorp mini II (Osaka, Japan).

### 2.4. Photocatalytic Activity Test

The photocatalytic activity of the as-prepared photocatalysts was examined using a batch reactor at room temperature. Diclofenac (DCF) solution with 5 mg/L concentration was prepared and used as a model pharmaceutical pollutant. The NaOH or HCl solutions separately, were used to adjust the pH value. The conical flask containing solution was placed inside a light reactor equipped with a 60 W LED lamp, positioned at 10 cm above the conical flask as a source of visible light. The photocatalytic activity test was conducted for two photocatalysts and for each test, a catalyst with the concentration of 1 g/L was dispersed in 100 mL of 5 mg/L DCF solution and it was stirred magnetically. The test was conducted at three different pH solutions (3, 7 and 9) to study the effect of solution pH on photocatalytic activity of photocatalysts. The effect of photolysis on DCF removal was evaluated by conducting the experiment without the photocatalyst. The solution samples (1.5 mL) were withdrawn and filtered by a 0.2 µm PTFE filter to remove the photocatalyst. The concentration of DCF was analyzed using Knauer high-performance liquid chromatography (HPLC) (C18 ODS reverse phase column, 250 × 4.6 × 5 micron with a UV-PDA detector at a wavelength of 278 nm, Shimadzu Corp., Kyoto, Japan) and acetonitrile solution was used as a mobile phase. The kinetic rate of DCF degradation was plotted using Ln(*C*/*C*_0_) versus time. The degradation percentage was calculated according to the following equation:$$Degradation\left(\%\right)=\frac{C_{0}-C_{e}}{C_{0}}\times 100$$ where *C*_0_ is the initial concentration and *C_e_* (mg/L) is the final concentration of DCF.

Prior to the photocatalytic degradation experiment under visible light, to reach an adsorption–desorption equilibrium between the DCF and the photocatalyst, the DCF solution containing the photocatalyst was stirred for 20 min in dark. Subsequently, the photocatalytic experiment was continued under visible light irradiation.

## 3. Results and Discussion

### 3.1. Structural and Chemical Characterization

#### 3.1.1. XRD

The phase structure and composition of the as-synthesized photocatalysts were characterized by X-ray diffraction (XRD). The XRD pattern of two photocatalysts is shown in Figure 1. Both photocatalysts, BAB1 and BAB2, contain tetragonal BiOCl, monoclinic sheelite BiVO_4,_ and cubic AgCl, demonstrating that the BiOCl/AgCl/BiVO_4_ heterojunction has been successfully constructed. The sharp peaks of the XRD patterns of both photocatalysts indicate their high crystallinity, while no additional impurity phases were found in the diffraction patterns. Furthermore, it was observed that with increased pH, the phase of BiVO_4_ changes and the overall intensity of the peaks is also reduced. The component phases with their corresponding reference code and score are presented in Table 1. As can be seen, the ratio of BiOCl to BiVO_4_ increased from 1.24 in BAB1 to 1.56 in BAB2.

#### 3.1.2. SEM and TEM

The SEM images of BAB1 and BAB2 heterojunctions are represented in Figure 2. BAB1 shows a full nanoplate morphology with the particle plate’s size between 100 and 500 nm, while BAB2 in addition to square nanoplates contains nanorods with a length between 100 and 200 nm (yellow square, Figure 2b). To further confirm the formation of the BiOCl/AgCl/BiVO_4_ heterojunction, high resolution transmission electron microscopy (HRTEM) images were obtained (Figure 2c,d) which confirms the observation obtained by SEM. 

The SAED patterns of the BAB1 and BAB2 photocatalyst (Figure 2e,f, respectively) show that all the phases are well-crystallized and exhibiting bright spots. The positions of the rings and spots (Figure 2e,f) indicate different planes of BiVO_4_, BiOCl, and AgCl, which are in agreement with the corresponding XRD data (Figure 1).

#### 3.1.3. UV-Vis Spectrophotometry

UV-vis DRS was recorded in the wave-length range of 200–1000 nm to characterize the optical properties of heterojunctions (BAB1 and BAB2), which is presented in Figure 3. As can be seen, there are two peaks in the spectra of both samples. The two peaks mean that there are two gaps—where the first one at higher wavelength corresponds to the optical gap, and the second one at lower wavelength corresponds to the fundamental gap [19].

The absorption edge of BAB1 is around 400 nm, while BAB2 has an absorption scale around 500 nm, with the steep shape spectrum which indicates interband transitions. Both BAB1 and BAB2 photocatalysts exhibit a shoulder in visible regions extending to 600 nm and 700 nm, respectively. This indicates that the BAB1 and BAB2 photocatalysts heterojunction can be used as excellent visible light photocatalysts. BiVO_4_ is an n-type semiconductor and BiOCl is regarded as a p-type semiconductor, coupling with an n-type semiconductor which along with AgCl forms a ternary n-p heterojunction, improving the visible light absorption and electron-hole separation [20].

The band gap was calculated with the plot of (F(R_∞_)*hν)^0.5^ against hν (eV). It was found that BAB1 has a band gap energy of 2.59 eV, while it was 1.74 for BAB2. This was calculated using the Kubelka-Munk function. The difference in the band gap value could refer to the difference in ratio of BiOCl to BiVO_4_.

#### 3.1.4. FTIR

Figure 4 shows the FT-IR spectra of BiOCl/AgCl/BiVO_4_ heterostructure nanocomposite, synthesized at pH = 1.2 and pH = 4, BAB1 and BAB2 respectively. For both photocatalysts, the peaks indicate the coexistence of BiOCl and BiVO_4_. The known peak at 512 cm^−1^ was observed in the spectra of both BAB1 and BAB2, which can be assigned to the Bi–O stretching vibration [6]. The peak around 606 cm^−1^ can be attributed to the bending vibration of the VO_4_^3−^, which is more significant in the BAB2 photocatalyst. The absorption bands at around 1200 cm^−1^ can also be assigned to the Bi–O stretching, which did not appear in sample BAB1. The absorption peak at 1460 cm^−1^ is assigned to the stretching vibration peak of the Bi–Cl band in the BiOCl structure [21]. The absorption bands of both samples suggest the presence of Bi–O and VO_4_^3−^ groups in the BiVO_4_/AgCl/BiOCl composites.

#### 3.1.5. BET

The nitrogen-adsorption-desorption isotherms of the samples were examined to find out the specific surface areas and pore size distribution at different synthesized pH values (Figure 5). For all samples, the isotherm is type IV Brunauer-Deming-Deming-Teller (BDDT classification), with very narrow hysteresis loops at high relative pressures, P/P_0_, indicating bimodal pore size distributions in the mesoporous region. Furthermore, high absorption at high P/P_0_ ranges indicates the formation of large mesopores. The shape of hysteresis loops is type H3, which related T to the aggregates of plate-like particles, giving rise to slit-like pores [22,23]. The type IV isotherm is also indicating T, the presence of mesopores with widths between 2 and 50 nm, and fine intra-aggregated micropores [24]. The pore size distribution (inset) calculated from the desorption branch of the nitrogen isotherm by the BJH (Barrett-Joyner-Halenda) method shows a very narrow range of pore diameter below ca. 4 nm for samples BAB1 and BAB2. Table 2 shows the BET surface area and texture properties of the samples. Generally, the organized porous structures are highly useful in photocatalysis, since they provide efficient transport of reactant and product molecules [25,26].

#### 3.1.6. Raman

Raman spectroscopy was used to examine the local structure of the synthesized polycrystalline photocatalysts powder. The Raman spectrum was recorded in the range of 0–2500 cm^−1^. Figure 6 shows the Raman spectra of the BiOCl/AgCl/BiVO_4_ heterostructure nanocomposite synthesized at pH = 1.2 and pH = 4, BAB1 and BAB2 respectively. For two samples, Raman spectra exhibit different behavior and these are the typical vibrational bands of BiVO_4_ and BiOCl. As seen in Figure 6, the BAB2 photocatalyst shows a dominant peak at 816 cm^−1^, which is ascribed to the V-O symmetric stretching bond of the tetrahedral BiVO_4_ [27,28]. The intensity of this band was higher in sample BAB1 comparing to BAB2, with a slight shift to the lower band. In the BAB1 sample, two peaks at 98 cm^−1^ and 131cm^−1^ can be ascribed to the A1g external and internal Bi–Cl stretching modes respectively, while the peak at 188 cm^−1^ was assigned to the Eg internal Bi–Cl stretching mode [29,30,31]. The Raman bands centered at 345 cm^−1^ represent the deformation mode of V–O tetrahedron vibrations of BiVO_4_ in photocatalyst BAB2. Similar vibration modes were detected for BAB2 as well, but with the main peak at 816 cm^−1^ with a different peak intensity. This confirms that in both samples, photocatalysts BiVO_4_ and BiOCl exist.

### 3.2. Photocatalytic Activity of BiOCl/AgCl/BiVO_4_ Photocatalysts

The photocatalytic activities of BAB1 and BAB2 photocatalysts were evaluated with respect to the degradation of DCF under visible light irradiation. Figure 7a shows adsorption capacity and the degradation efficiencies of DCF by BAB1 and BAB2 photocatalysts. It is evident that both catalysts had negligible DCF adsorption. However, BAB1 exhibits higher photocatalytic activity with a degradation efficiency of 72%, comparing with BAB2’s rate of 47% under visible light irradiation for 2.5 h. Results shown in Figure 7b represent the corresponding kinetics for DCF degradation by BAB1 and BAB2. As is presented in Figure 7b, the BAB1 catalyst had the higher correlation coefficient and rate constant. The photocatalytic activity of BAB2 was lower than BAB2 activity, which can be due to both the lower surface area and specific phase ratio of the mentioned photocatalysts.

The effect of pH value on the degradation rate of DCF was studied under 3 conditions (pH = 3.7 and 9). The highest degradation rate was observed in acidic conditions (pH = 3), which was very close to the results of neutral condition at pH = 7 (Figure 7c). This pH-dependent degradation has been explained based on the presence of two major different DCF species (DCF neutral form and anionic form, and this pH dependency can be related to the acid-base speciation distribution of DCF [32].

### 3.3. Brief Comparison of Bi-Based Photocatalysts

Recently, Bi-based photocatalysts and bismuth oxyhalides (BiOX, X = F, Cl, Br and I) have received attention as the visible light active photocatalysts. As performed in other studies, various strategies such as metal and/or non-metal doping, heterojunction formation, inner coupling between different BiOX photocatalyst, enhancement in Bi content, and use of sensitizers have been used to improve the photocatalytic performance of Bi-based photocatalysts [33]. Dy-doped BiOCl reported higher photocatalytic activity as compared to pure BiOCl for rodamine B degradation [34]. A visible-light-active BiOCl/BiVO_4_ p-n heterojunction photocatalyst was synthesized by He et al. [35]; this could degrade methyl orange completely within 600 min of reaction time, and showed higher efficiency in comparison to pure BiVO_4_ and BiOCl. Ma et al. [36] reported the preparation of mesoporous spindle-like BiVO_4_/BiOCl nanocomposite and used it for norfloxacin degradation, which reached complete removal of it in 1 h visible light irradiation. In another study, highly efficient BiVO_4_/BiOCl {001} heterojunctions—via in situ chemical transformation and its performance—were evaluated by photodegradation of Rhodamine B (RhB) using a 500 W xenon lamp [37]. Under an initial concentration of RhB 20 mg/L, more than 95% RhB degradation was achieved. However, Bi-based photocatalysts—especially different BiOX couples or heterojunction photocatalysts—are highly active. In practical application view, a photocatalyst with a less complicated synthesis method and low energy consumption was introduced in both the processes of synthesis and treatment. Liu and his colleagues [38] synthesized BiOI/BiOCl heterojunction structure and reported 78% of dye removal after 150 min. Zhang et al. reported a nono-composite such as Ag_3_PO_4_/g_C_3_N_4_ for 100% removal of DFC, the concentration of which was only 1 mg/L only [39] comparing to the concentration used in this study which used 5 mg/L.

However, in our study we introduced a new heterojunction of three compounds (BiOCl/AgCl/BiVO_4_) in one pot which could achieve 72% DCF degradation (10 mg/L) under visible light of a 60 W LED lamp. Comparing to other Bi based categories of photocatalysts, BiOCl/AgCl/BiVO_4_ apart from introducing a new heterojunction, introduces a simple and one-step 4 h synthesis method to produce the photocatalyst which can control the ratio of the semiconductors in composite by pH adjustment. 

## 4. Conclusions

In summary, the p-n BiOCl/AgCl/BiVO_4_ heterojunction was synthesized for the first time via a one-pot hydrothermal process, which was performed at two different pH values. The obtained BiOCl/AgCl/BiVO_4_ heterojunction at pH = 1.2 exhibits higher photocatalytic activity for degradation of diclofenac than the one synthesized at pH = 4. This study offers a novel one-pot preparation of high-performance ternary heterojunction materials as photocatalysts, which can be used for solar harvesting and the photo-catalytic degradation of organic pollutants.

The change in pH during synthesis influenced morphology and the increasing pH led to the formation of nano-rods in addition to the nono-sheets. The morphology and surface structure of as-obtained heterostructured BiOCl/AgCl/BiVO_4_ were confirmed using SEM and TEM. 

It was also found that the XRD BiVO_4_/BiOCl ratio was higher for BAB1, which agreed with the Raman results. This was an important role, leading to a remarkable enhancement of the visible light photocatalytic degradation of DCF. This proves that when keeping the synthesis condition constant, changing the pH can lead to a change in the ratio of BiVO_4_ to BiOCl, and consequently a change in the photocatalytic activity rate of the heterostructure photocatalyst. 

This work offers a simpler way of fabricating and designing nanostructures with an appropriate ratio of semiconductors involving the compound, by pH adjustment during synthesis. This is a feasible approach for developing highly visible-light-active heterojunction photocatalysts. The results of degradation also suggest that this photocatalyst is an excellent candidate for the degradation of pharmaceutical/emerging pollutants.

## Figures and Tables

**Figure 1 materials-12-02297-f001:**
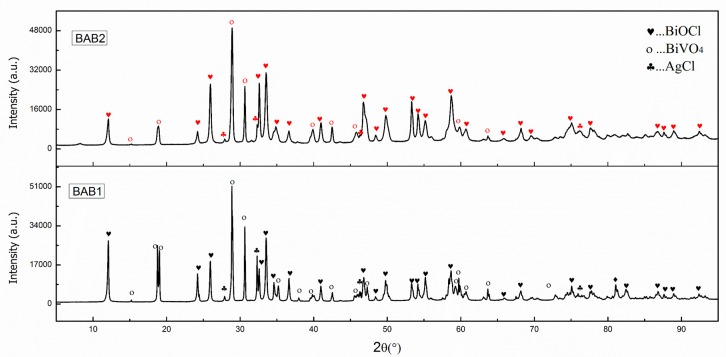
XRD patterns of BAB1 and BAB2 photocatalyst.

**Figure 2 materials-12-02297-f002:**
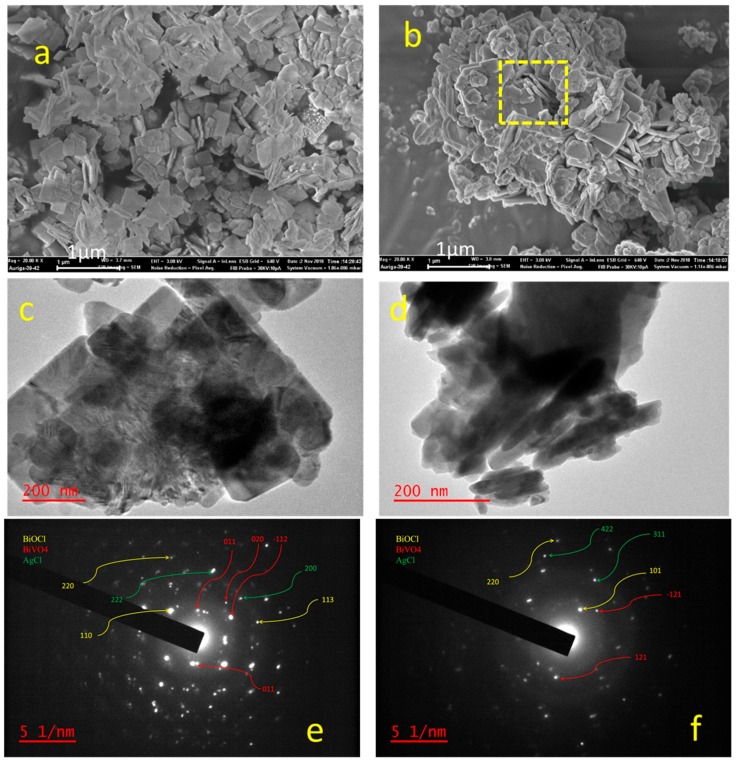
SEM image of BAB1 (**a**), BAB2 (**b**), TEM image of BAB1 (**c**), BAB2 (**d**) and SAED pattern of a BAB 1 (**e**) and BAB2 (**f**).

**Figure 3 materials-12-02297-f003:**
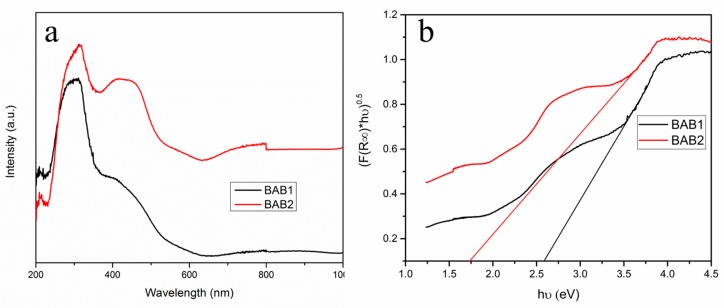
(**a**) Ultraviolet–visible diffuse reflectance spectra (UV-Vis DRS) and (**b**) determination of indirect interband transition energies for BAB1 and BAB2.

**Figure 4 materials-12-02297-f004:**
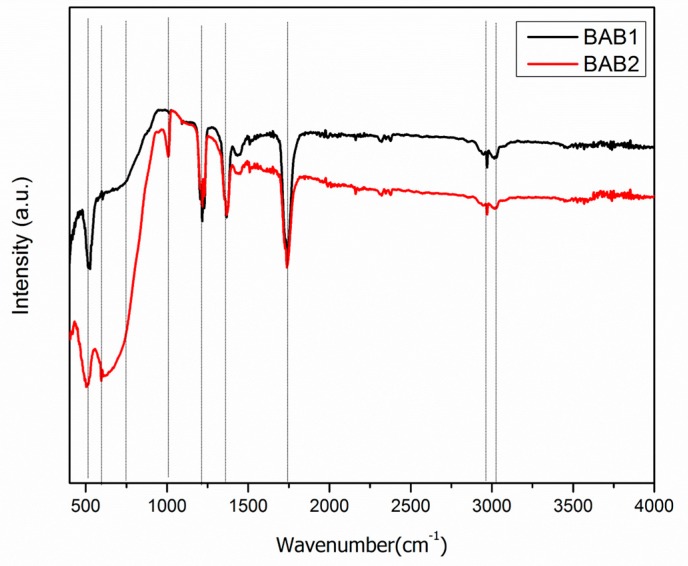
FT-IR spectra of BiOCl/AgCl/BiVO_4_ synthesized at pH = 1.2 (BAB1) and BiOCl/AgCl/BiVO_4_ synthesized at pH = 4 (BAB2).

**Figure 5 materials-12-02297-f005:**
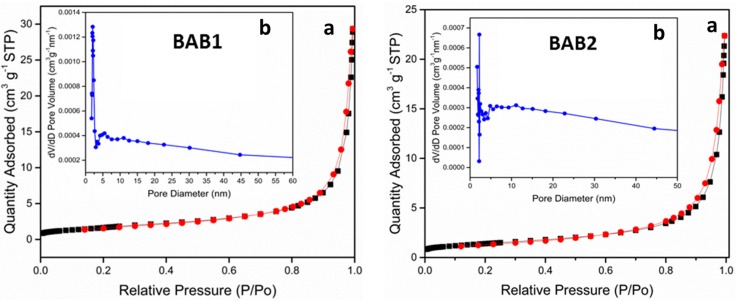
(**a**) N_2_-sorption isotherms and (**b**) corresponding pore-size distribution (inset) curves for BiOCl/AgCl/BiVO_4_ synthesized at pH = 1.2 (BAB1) and BiOCl/AgCl/BiVO_4_ synthesized at pH = 4 (BAB2).

**Figure 6 materials-12-02297-f006:**
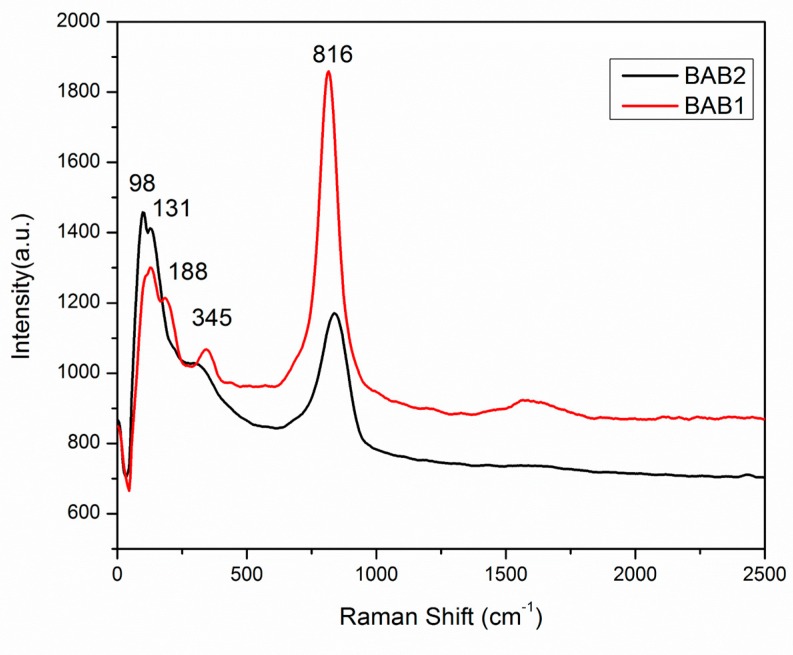
Raman spectra of the BiOCl/AgCl/BiVO_4_ synthesized at pH = 1.2 (BAB1) and BiOCl/AgCl/BiVO_4_ synthesized at pH = 4 (BAB2).

**Figure 7 materials-12-02297-f007:**
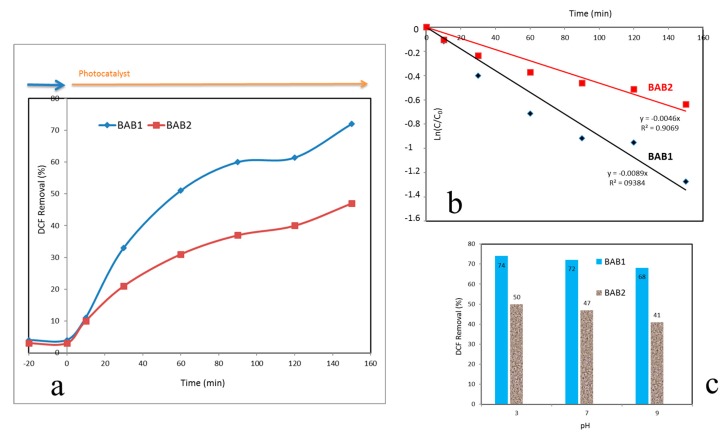
Photocatalytic degradation of degradation of diclofenac (DCF) (**a**); the corresponding pseudo-first-order reaction kinetics (**b**); and corresponding photocatalytic degradation efficiencies in solutions with a different pH (**c**) for BAB1 and BAB2 photocatalysts.

**Table 1 materials-12-02297-t001:** Pattern list of XRD patterns of BAB1 and BAB2 photocatalyst.

Compound	BAB1		BAB2	
	Reference Code	Score	Reference Code	Score
BiOCl	04-007-4915	62	04-007-4915	75
BiVO_4_	04-016-0302	50	00-014-0688	48
AgCl	01-071-5209	31	04-007-3906	33

**Table 2 materials-12-02297-t002:** BET surface areas, pore volume, and pore size in the samples.

Sample	S_BET_ (m^2^/g) ^a^	Pore Volume (cm^3^/g) ^b^	Average Pore Size (nm) ^c^
BAB1	6.16	0.044	25.9
BAB2	5.09	0.033	23.8

^a^ BET surface area calculated from the linear part of the BET plot. ^b^ Barrett-Joyner-Halenda (BJH) adsorption cumulative volume of pores between 17.0 Å and 3000.0 Å diameter. ^c^ Adsorption average pore diameter (4V/A by BET).

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
