# Peer review of "One-Pot Synthesized Visible Light-Driven BiOCl/AgCl/BiVO4 n-p Heterojunction for Photocatalytic Degradation of Pharmaceutical Pollutants"

_materials, 2019, doi:10.3390/ma12142297_

Round 1

Reviewer 1 Report

In this article, the authors synthesized BiOCl/AgCl/BiVO4 ternary composites with enhanced visible light absorption which was obtained through one-pot hydrothermal method at two different pHs. The obtained BiOCl/AgCl/BiVO4 ternary composites were evaluated for the degradation of diclofenac (DFC) under visible light illumination.

How to prove the existence of heterojunction mentioned in this article?

In UV-DRS spectral analysis, the BiOCl/AgCl/BiVO4 (pH = 4) composite showed higher light absorption in visible region when compared with BiOCl/AgCl/BiVO4 (pH = 1.2) composite. However, the BiOCl/AgCl/BiVO4 (pH = 1.2) composite showed higher degradation efficiency under visible light illumination. Why?

What is the band gap energy of BiOCl/AgCl/BiVO4 ternary composite?

Why did the photocatalytic activity decrease at pH 7 and 9?

This manuscript exhibits a similarity percentage higher than 30% with already published papers, which should be lowered.

The authors should provide the possible photocatalytic degradation mechanism of DFC over BiOCl/AgCl/BiVO4 under irradiation.

Author Response

In this article, the authors synthesized BiOCl/AgCl/BiVO4 ternary composites with enhanced visible light absorption which was obtained through one-pot hydrothermal method at two different pHs. The obtained BiOCl/AgCl/BiVO4 ternary composites were evaluated for the degradation of diclofenac (DFC) under visible light illumination.

How to prove the existence of heterojunction mentioned in this article?

From the HRTEM images, we can see that the lattice fringes of the two semiconductor materials that are interconnected with each other which is indicating that hetero-junctions are formed between.

In UV-DRS spectral analysis, the BiOCl/AgCl/BiVO4 (pH = 4) composite showed higher light absorption in visible region when compared with BiOCl/AgCl/BiVO4 (pH = 1.2) composite. However, the BiOCl/AgCl/BiVO4 (pH = 1.2) composite showed higher degradation efficiency under visible light illumination. Why?

In the photocatalytic activity the visible light absorption and consequently band gap narrowing is one of the parameters effecting the photocatalyst activity, as electron-hole recombination also play an important role in photocatalytic activity as the electron-hole recombination rate increases the chances of photocatalytic activity reduces, which could be the reason in this work. Higher surface area in BAB1 also can be another advantage in higher catalytic activity.

What is the band gap energy of BiOCl/AgCl/BiVO4 ternary composite?

Than band gap energy of both samples prepared have been calculated and have been added to the manuscript.

Why did the photocatalytic activity decrease at pH 7 and 9?

This pH dependent degradation has been explained based on presence of two major different DCF species (neutral form and anionic form and this pH dependent can be related to the acid-base speciation distribution of DCF (Chen et al., 2015). This explanation has been added in the manuscript as well.

This manuscript exhibits a similarity percentage higher than 30% with already published papers, which should be lowered.

The manuscript has been revised accordingly to take care of similarity.

The authors should provide the possible photocatalytic degradation mechanism of DFC over BiOCl/AgCl/BiVO4 under irradiation.

It is evident that both catalysts had negligible DFC in dark adsorption experiment revealing that the main DFC removal is due to the visible light photocatalytic activity. In addition the hetrostructure of photocatalyst could help to efficient electron-hole separation. The ratio of BiOCl/BiVO4 was lower in BAB1 which could be another key parameter clearing the mechanism of the photocatalyst.

Reviewer 2 Report

The work presented by the authors is well strcutured and presented; I think some minor revisions are necessary prior to publication.

In Figure 1, it is difficult to distinguish between the different symbols belonging to the different phases; I suggest the authors use symbols in different colours or choose symbols which are more clearly distinguishable (i.e. hollow circles, etc.).

In Figure 2a, images with higher magnification should be provided, to see better the morphology of the materials.

Finally, regarding the photocatalytic activity, some comparison with literature data should be added.

Author Response

The work presented by the authors is well strcutured and presented; I think some minor revisions are necessary prior to publication.

In Figure 1, it is difficult to distinguish between the different symbols belonging to the different phases; I suggest the authors use symbols in different colours or choose symbols which are more clearly distinguishable (i.e. hollow circles, etc.).

Response: The symbols have been changed to different colours/style and now they are clearly distinguishable.

In Figure 2a, images with higher magnification should be provided, to see better the morphology of the materials.

The scale in the scale bar has been magnified.

Finally, regarding the photocatalytic activity, some comparison with literature data should be added.

The main idea of this work is being able to create a novel BiOCl/AgCl/BiVO4 heterojunction in one-pot synthesis and report for the first time and further successful modification of the catalyst with a parameter such pH to change the ratio of different phases. The diclofenac has been chosen as a model pollutant to evaluate the photocatalyst activity.

Round 2

Reviewer 1 Report

The authors should take care of the similarity of the manuscript seriously. I do find many similiarities in this manuscript, which has not been improved yet. Furthermore, the comments raised by the reviewer have not been solved clearly.

Author Response

We have carefully and seriously taken the editor and reviewers’ comments into account, and corresponding revisions have been made to improve our original manuscript.